# Review of State-of-the-Art Green Monopropellants: For Propulsion Systems Analysts and Designers

**Ahmed E. S. Nosseir** [1,2,*], **Angelo Cervone** [1,*] and **Angelo Pasini** [2]

1   Department of Space Engineering, Faculty of Aerospace Engineering, Delft University of Technology (TU Delft), 2629 HS Delft, The Netherlands
2   Sede di Ingegneria Aerospaziale, Dipartimento di Ingegneria Civile e Industriale, Università di Pisa (UniPi), 56122 Pisa, Italy; angelo.pasini@unipi.it
*   Correspondence: a.nosseir@studenti.unipi.it or a.e.s.nosseir@student.tudelft.nl (A.E.S.N.); a.cervone@tudelft.nl (A.C.)

**Abstract:** Current research trends have advanced the use of "green propellants" on a wide scale for spacecraft in various space missions; mainly for environmental sustainability and safety concerns. Small satellites, particularly micro and nanosatellites, evolved from passive planetary-orbiting to being able to perform active orbital operations that may require high-thrust impulsive capabilities. Thus, onboard primary and auxiliary propulsion systems capable of performing such orbital operations are required. Novelty in primary propulsion systems design calls for specific attention to miniaturization, which can be achieved, along the above-mentioned orbital transfer capabilities, by utilizing green monopropellants due to their relative high performance together with simplicity, and better storability when compared to gaseous and bi-propellants, especially for miniaturized systems. Owing to the ongoing rapid research activities in the green-propulsion field, it was necessary to extensively study and collect various data of green monopropellants properties and performance that would further assist analysts and designers in the research and development of liquid propulsion systems. This review traces the history and origins of green monopropellants and after intensive study of physicochemical properties of such propellants it was possible to classify green monopropellants to three main classes: Energetic Ionic Liquids (EILs), Liquid NOx Monopropellants, and Hydrogen Peroxide Aqueous Solutions (HPAS). Further, the tabulated data and performance comparisons will provide substantial assistance in using analysis tools—such as: Rocket Propulsion Analysis (RPA) and NASA CEA—for engineers and scientists dealing with chemical propulsion systems analysis and design. Some applications of green monopropellants were discussed through different propulsion systems configurations such as: multi-mode, dual mode, and combined chemical–electric propulsion. Although the in-space demonstrated EILs (i.e., AF-M315E and LMP-103S) are widely proposed and utilized in many space applications, the investigation transpired that NOx fuel blends possess the highest performance, while HPAS yield the lowest performance even compared to hydrazine.

**Keywords:** green propellant; monopropellant; chemical rocket propulsion; small satellites; CubeSat; in-space propulsion; liquid propulsion system

## 1. Introduction

The current trend in the rocket propulsion field is directed towards greenifying the use of propellants. Monopropellant hydrazine was classically widely used and favored for thrusters and gas generators due to its high performance, system's simpler design, and "clean" relatively cool exhaust products as compared to bipropellant systems at that time [1]. European CHemicals Agency (ECHA) in (REACH) Registration, Evaluation, Authorization and restriction of Chemicals has included hydrazine on the list of Substances of Very High Concern (SVHC) for authorization, thus opening a process that will eventually lead to a ban on the use of hydrazine and its derivatives as space propellant in European countries [2]. Moreover, transportability and handling of hydrazine and similar hazardous propellants

extend an economic burden on the space industry. Accordingly, greener alternatives that would compensate for these drawbacks are being studied and developed rapidly nowadays [3,4]. Different global entities were involved in accelerating such research activities through various projects and missions such as Green Advanced Space Propulsion (GRASP), Pulsed Chemical Rocket with Green High Performance Propellants (PulCheR), and Replacement of hydrazine for orbital and launcher propulsion systems (RHEFORM) European projects and Green Propellant Infusion Mission (GPIM) technology demonstrator project by NASA. Since the beginning of their development, modern green propellants have shown high favorability not only in terms of operability, cost efficiency, and environmental safety but also in performance, and physicochemical properties [5].

As widely interpreted, green propellants are defined as low-hazard, low-toxicity, environmentally friendly propellants during various phases of spacecraft development, launch, and operations. Such propellants provide safe handling and storability when compared to conventional toxic propellants such as hydrazine and its derivatives that require special handling protocols and adhering to strict safety measurements that, in addition to others, include using Self-Contained Atmospheric Protective Ensemble (SCAPE) suits. Due to their favorable characteristics, green propellants demonstrate higher commercial value by being able to cut costs related to transportation, storage, handling, and further reduces ground operations time. Recently, a more specified definition has been noted by Mayer et al. [3], based on the Acute Toxicity Classification (ATC) by the Global Harmonized System of classification and labeling of chemicals (GHS) [6], which denotes that propellants possessing ATC levels of three and safer are considered as green propellants. ATC levels are typically categorized on a 1:5 scale where level one denotes the most toxic class and level five is considered the least toxic class.

Before discussing each green monopropellant family in detail, relatively different classifications in literature were proposed for green alternatives of hydrazine monopropellant, one of them considered mainly three types: Energetic Ionic Liquids, Hydrogen Peroxide, and Nitrous Oxide by Batonneau et al. [7] as cited in [8]. However, Mayer et al. [3] further classified propellants including nitrogen compounds with oxygen into two groups: *Oxides of Nitrogen* subcategory and *Nitro Compounds* subcategory. The former included mono and dinitrogen oxides (NO, $NO_2$, $N_2O$, $N_2O_3$, $N_2O_4$, $N_2O_5$), which were evaluated as potential oxidizers for bipropellant systems; among which the only compound that was considered as potential green propellant was the nitrous oxide ($N_2O$) due to its relative nontoxicity (GHS [6] class 5) and being liquid within a wide part of the typically requested temperature-pressure envelope of [−30, +80] °C and [0.1, 3] MPa, respectively. The latter, *Nitro Compounds* group, was described as organic substances containing dinitrogen monoxide (i.e., hydrocarbons and nitrous oxide) group, such as (mono)nitromethane ($CH_3NO_2$ or NM), which was also considered as a promising candidate green propellant being relatively nontoxic (GHS class 4) [3]. Gohardani et al. [9] aimed at reviewing and investigating a number of promising green propellants in a part of the paper; the mentioned three candidate propellant categories were hydrogen peroxide, nitrous oxide fuel blends, and ionic liquids. The former two were rather qualitatively described, while the latter was further discussed in quantitative data introducing only two types of green monopropellants.

This review article will be classifying green monopropellants into three more collective major categories:

1. Energetic Ionic Liquids (EILs) (or premixed oxidizer/fuel ionic aqueous solutions).
2. Liquid NOx Monopropellants (either in binary compound, nitro compound, or premixed/blend form).
3. Hydrogen Peroxide Aqueous Solutions (HPAS).

The main concern of the discussed propellants is to be used for green in-space propulsion, either primary or auxiliary (secondary) propulsion, in a wide range of spacecrafts starting from small satellites to upper-stages (kick-stages) of launch vehicles, with the capability of providing continuous or pulsed low- to high-thrust in a range of millinewtons up to 600 N. Few applications for some of the discussed green monopropellants in various

propulsion system configurations were introduced and schematized. An emphasis was made on the definitions of some common terms describing such configurations, which are: multi-mode, combined chemical-electric, and dual mode propulsion, and the differences between each were elaborated.

Rocket Propulsion Analysis (RPA) and NASA Chemical Equilibrium Analysis (CEA) are important analysis tools for the conceptualization and design of chemical rocket engines. These analysis tools are widely used in academia, especially by young scientists and engineers for the analysis and preliminary design of propulsion systems, due to their maturity, user friendliness, and free availability. Engine performance analyses are provided by these tools, which are able to calculate theoretical (ideal) rocket performance, estimate delivered performance values, study combustion products, along with nozzle design optimization and engine mass estimation. Concerning green propellants representation, these tools have limitations in providing enough green propellants formulations in their database. Fortunately, it is allowable to formulate any propellant of interest by providing sufficient data, such as the exact composition ratio and thermochemical properties of constituents of sought propellant. This review article collected and presents these necessary data for a wide range of state-of-the-art green monopropellants from relevant literature sources with highest reliability. It is worth noting that, for any non-proprietary green monopropellant mentioned in this review, it will be possible to integrate the provided formula composition and thermochemical data of the constituents to perform the required analyses. As for proprietary propellants, sufficient thermochemical data on all the constituents is provided along with the overall thermodynamic and physical properties of such propellants; however, the exact constituents ratio (i.e., weight %) is not available.

## 2. Green Monopropellants Classification

A controversial topic arises when referring to some modern green propellants, whether to address them by the term "monopropellants" or by more specific terms including (premixed propellants, fuel blends, or mixtures). Monopropellants are commonly defined as propellants consisting of chemical compounds (for example $N_2H_4$), which release energy through exothermic chemical decomposition. Since the evolution of liquid gun propellants based on HAN compound and other nitrate salts aqueous solutions—discussed in the next section, the term "monopropellants" was used to describe such premixed formulations. As widely used in literature and industry, some modern green propellants, for instance the EILs, are undoubtedly classified and described as "monopropellants." Basically, it can be interpreted from the previous example that a propellant that is stored in a single tank and is able to decompose from its storage form by the help of a catalyst or other ignition method, such as thermal or electric ignition, can be considered a "monopropellant" as long as it does not require another separately stored propellant for decomposition. Nitrous Oxide Fuel Blends (NOFBs)—discussed in Section 2.2—were mostly described as green monopropellants as well, maintaining the above-mentioned unique conditions for storage and decomposition of monopropellants.

This section handles each of the three classes proposed for the state-of-the-art green monopropellants. History and origins of development are entitled along with technical data and characteristics including the chemical formulations and constituents of each monopropellant. Thermodynamic and thermochemical properties are gathered from various literature sources—especially resources solely focused on studying propellants thermochemistry. Flight heritage of mature propellants is mentioned, and promising lab test results of other promising propellants are highlighted when possible. Ignition techniques are another important property sought by propulsion systems designers that assist in giving insights about spacecraft mass and volume preliminary requirements as well as electric power needs, thus is noted and discussed deliberately whenever reliable data were available. Finally, physical properties and performance parameters of each propellant are tabulated, and relevant group comparisons are made for the reader's convenience.

*2.1. Energetic Ionic Liquids (EILs)*

Energetic Ionic Liquids (or premixed oxidizer/fuel ionic propellant blends) consist of oxidizer salts dissolved in aqueous solutions, called Ionic Liquids (ILs), mixed with Ionic Fuel (IF) or Molecular Fuel (MF), refer to Table 1, forming a premixed propellant (i.e., Energetic Ionic Liquid monopropellant as widely referred to among the rocket propulsion community [8]). Addition of the fuel component increases the performance of the propellant blend by reducing the high adiabatic temperature of the ionic liquid binary aqueous solution and further stabilizing the combustion process. Typically, methanol is used to control the burning rate of the monopropellant while the ammonium nitrate (AN) is used as a stabilizer [10] beside other stabilizing additives; a new article by M. Claßen et al. [11] introduced novel additive promoters of new azido esters and suggested it would improve the total energy and performance of ionic liquid propellants. As an example, the maximum specific impulse of 78 wt% ADN in water mixture (Ionic Liquid) is 192 s when used as monopropellant, while the specific impulse rises to 252 s when methanol (molecular fuel) is added to the mixture, as in the case of LMP-103S (63.4 wt% ADN, 25.4 wt% water and 11.2 wt% methanol) at a nozzle area expansion ratio of 50 [12] as cited in [3]. In the next paragraphs different EILs will be reviewed (i.e., HAN, HAN/HN, HNF, ADN) based on green monopropellants emphasizing on their composition, physical properties, performance, stability of storage and handling, toxicity, material compatibility, ignition methods, and in-flight heritage or proposed missions.

HAN-based monopropellants origins can be traced back to the development of liquid gun propellants (LGPs) in the U.S. Army [13]. Three formulations of LGPs were addressed, namely LP1846, LP1845, and LP1898 [5] and their properties are listed in Table 2. The first two of these aqueous solutions are HAN/TEAN-based (tri-ethanol-ammonium nitrate) while the third is HAN/DEHAN-based (di-ethyl hydroxyl ammonium nitrate). The unsuitability of these propellants for rocket's relatively low combustion pressure [14], as well as the high combustion temperature (2500 K [5]) eventually lead to the development of the state-of-the-art AF-M315E (Air Force Monopropellant 315E) HAN-based green monopropellant for space propulsion formulated by the U.S. Air Force Research Laboratory (AFRL) [15].

AF-M315E when decomposed produces an adiabatic flame temperature around 2100 K, which is much higher than that of hydrazine (nearly 1200 K). AF-M315E offers 13% increase of specific impulse and 63% increase in density over hydrazine [13], which makes it superior in the miniaturization of propulsion systems over the latter. The propellant possesses high solubility and negligible vapor-pressure of all its solution constituents, thus promoting low toxicity hazards and high mixture stability at various temperature levels, which makes exposure in open environments have no safety issues [30]. An advantage AF-M315E possesses over current state-of-the-art green propellants is its maturity. Thorough development has taken place to reach this product and be able to test in space on 1 N and 22 N thrusters through the Green Propellant Infusion Mission (GPIM) launched in 2019 [31]. However, a disadvantage over the latest state-of-the-art green propellants rises from the relatively high flame temperature, which makes it difficult to rapidly manufacture an economic and simpler design of thrusters especially for the micro/nano satellites industry. It is worthy of mentioning that current advancements, especially related to rapid prototyping, in low-cost thrusters of small spacecrafts would benefit greatly from using additive manufacturing techniques such as metal 3D printing. Such techniques facilitate the design process and reduce the build time, they typically use metal alloys such as Ti-6Al-4V (Ti64) and Inconel®-625 (nickel-chromium superalloy) with melting points of approximately 1900 and 1570 K, respectively [32,33]. Catalytic decomposition of AF-M315E requires higher preheating temperature, compared to hydrazine, where it typically consumes up to 15 kJ of energy [34] and the catalyst bed preheating nominal start temperature is 315 °C [35], while Busek Co. Inc researchers reported successful ignition at 400 °C preheating temperature [13].

**Table 1.** Energetic ionic liquids: oxidizers and fuels thermochemical properties [8,16–19].

| Ionic Oxidizer | | Molecular Weight (g mol$^{-1}$) | Standard Heat of Formation (kJ mol$^{-1}$) |
|---|---|---|---|
| HAN, hydroxyl ammonium nitrate | [NH$_3$OH]$^+$[NO$_3$]$^-$ | 96.04 | −338.97 [20] |
| ADN, ammonium dinitramide | [NH$_4$]$^+$ [N(NO$_2$)$_2$]$^-$ | 124.06 | −134.6 [21] as cited in [22] |
| HNF, hydrazinium nitroformate | [N$_2$H$_5$]$^+$ [C(NO$_2$)$_3$]$^-$ | 183.08 | −72.104 [20] |
| AN, ammonium nitrate | [NH$_4$]$^+$ [NO$_3$]$^-$ | 80.043 | −365.28 [20] |
| HN, hydrazinium nitrate | [N$_2$H$_5$]$^+$ [NO$_3$]$^-$ | 95.06 | −211.36 [20] |
| **Ionic Fuel** | | | |
| AA, ammonium azide | [NH$_4$]$^+$ [N$_3$]$^-$ | 60.06 | 113.66 [20] |
| HA, hydrazinium azide | [N$_2$H$_5$]$^+$ [N$_3$]$^-$ | 75.07 | 228.53 [20] |
| HEHN,2-hydroxyethyl-hydrazinium nitrate | [HO-C$_2$H$_4$-N$_2$H$_4$]$^+$ [NO$_3$]$^-$ | 139.11 [23] | −388.69 [24] |
| **Molecular Fuel** | | | |
| MMF, mono-methylformamide | CH$_3$HNCHO | 59.067 | −247.4 [22] |
| DMF, di-methylformamide | (CH$_3$)$_2$NCHO | 73.094 | −239.3 [25] as cited in [22] |
| Methanol | CH$_3$OH | 32.04 | −238.77 [20] |
| Ethanol | CH$_3$CH$_2$OH | 46.07 | −277.755 [20] |
| Glycerol | (CH$_2$OH)$_2$CHOH | 92.094 | −669.6 [26] |
| Glycine | NH$_2$CH$_2$COOH | 75.07 | −528.0 [27] |
| Urea | CO(NH$_2$)$_2$ | 60.06 | −333.43 [20] |

**Table 2.** Composition of US Army liquid gun propellants (LGPs) [28,29].

| Propellant | Component, wt% | | | |
| | HAN [NH$_3$OH]$^+$ [NO$_3$]$^-$ | TEAN [NH(C$_2$H$_4$OH)$_3$]$^+$ [NO$_3$]$^-$ | DEHAN [(CH$_3$CH$_2$)HNOH]$^+$ [NO$_3$]$^-$ | Water H$_2$O |
|---|---|---|---|---|
| LP1846 | 60.8% | 19.2% | 0.0% | 20.0% |
| LP1845 | 63.2% | 20.0% | 0.0% | 16.8% |
| LP1898 | 60.7% | 0.0% | 19.3% | 20.0% |

SHP163 is another very interesting HAN-based green propellant, which was being developed since the year 2000 at the Institute of Space and Astronautical Science (ISAS)/(JAXA). SHP163 is composed of 73.6 wt% HAN, 3.9 wt% AN, 16.3 wt% methanol, and 6.2 wt% water [36]. This propellant has density of 1.4 g cm$^{-3}$ yet achieves high volumetric specific impulse $\rho I_{sp} = 396$ g s cm$^{-3}$, which is higher than AF-315E (at 0.7 MPa chamber pressure and 50:1 nozzle expansion ratio at frozen conditions) [36]. The flame temperature is considered very high, as it records about 2400 K [36,37]. As SHP163 shows to be one of the most energetic propellants for use in a thruster, it demonstrates operational stability and shows enough safety levels to be accepted as a green and safe liquid propellant [38]. SHP163 is only ignitable using a preheated catalyst-bed under 1.0 MPa [39,40]. Finally, SHP163 was tested in space in the Green Propellant Reaction Control System (GPRCS) utilizing a 1 N class thruster in the RAPIS-1 satellite launched in 2019 by JAXA.

HNPxxx family (High-performance Non-detonating Propellant) are HAN/HN-based green propellants that have been under development for over 10 years by IHI Aerospace co. in Japan. This green monopropellant family include HNP209, HNP221, and HNP225, and they are formulated from HAN, HN, methanol, and water [33]. They all possess volume specific impulse $(\rho I_{sp})$ superior to hydrazine, but what characterizes them most is their relatively low adiabatic flame temperature compared to other energetic ionic liquid monopropellants such as AF-M315E and SHP163. HNP209 typically has a theoretical specific impulse around 260 s with the highest combustion temperature (~1900 K), while HNP221 and HNP225 have specific impulse of 241 and 213 s, respectively (at chamber pressure of 1.0 MPa and expansion ratio of 100:1) [41–43], as shown in Table 3.

**Table 3.** Physical properties and performance (@ 1 MPa chamber pressure, 100:1 expansion ratio, and vacuum conditions, using NASA Chemical Equilibrium Analysis (CEA) [32] and verified from [4,30,33,36,37,44–46]).

| Properties | Hydrazine | AF-M315E | SHP163 | HAN/HN-Based | |
| --- | --- | --- | --- | --- | --- |
| | | | | HNP221 | HNP225 |
| Theoretical Specific Impulse $I_{sp}$ (s) | 239 | 260–270 | 276 | 241 | 213 |
| Density $\rho$ (g cm$^{-3}$) (@ 20 °C) | 1.0 | 1.47 | 1.4 | 1.22 | 1.16 |
| Volumetric Specific Impulse $\rho I_{sp}$ (g s cm$^{-3}$) | 239 | ~390 | 386 | 294 | 247 |
| Adiabatic Flame Temperature (K) | 1170 | 2166 | 2401 | 1394 | 990 |
| Freezing Point (°C) | 1.5 | $<-80$ | $\leq -30$ | $\leq 0$ | $\leq -10$ |

HNP225 is the one among the family with the least adiabatic flame temperature around 1000 K (even less than hydrazine ~1200 K), while HNP221 is approximately 1400 K [33,43]. The low temperature combustion gasses allowed IHI Aerospace co. to develop low-cost thrusters since the need for high heat resistant materials or complex cooling for the thruster's combustion chamber is no longer required. The HNP2xx family of propellants are ignited using catalytic decomposition. Igarashi et al. 2017 [33] performed tests for HNP221 and HNP225 with newly developed catalysts showing excellent response and combustion pressure stability compared to hydrazine, either in continuous mode or pulsed mode operation, with preheating temperatures starting from 200 and 300 °C for HNP221 and HNP225, respectively.

GEM or the Green Electrical Monopropellant is a novel HAN-based energetic ionic liquid composed of HAN, AN, (2,2′-dipyridyl), (1,2,4-triazole), 1H-pyrozol, and water [47]. GEM is a proprietary of Digital Solid-State Propulsion company (DSSP) [47] and is developed as a superior replacement for AF-M315E [46]. This propellant is demonstrated on a lab-scale to be capable of taking place in a multi-mode propulsion system. A "multi-mode" system is where a propulsion system in a satellite can operate as two or more separate modes (e.g., chemical high-thrust mode and electric high-specific impulse mode) under a condition of using a shared propellant tank [48] as cited in [49]. The most appealing properties in GEM is that it can also be electrically ignited without the use of any heavy catalytic beds, and it possesses a significantly large volumetric specific impulse as compared to AF-M315E and the ADN-based LMP-103S green monopropellants [46], refer to Table 4.

**Table 4.** Performance and physical properties of Green Electric Monopropellant (GEM) compared to state-of-the-art Green Monopropellants (@ 2.0 MPa chamber pressure, 50:1 expansion ratio, and vacuum conditions) [46].

| Properties | Hydrazine | LMP-103S | AF-M315E | GEM |
|---|---|---|---|---|
| Theoretical Specific Impulse $I_{sp}$ (s) | 236 | 252 | 266 | 283 |
| Density $\rho$ (g cm$^{-3}$) (@ 20 °C) | 1.0 | 1.24 | 1.47 | 1.51 |
| Volumetric Specific Impulse $\rho I_{sp}$ (g s cm$^{-3}$) | 236 | 312.48 | 391 | 427 |
| Vapor Pressure $P_V$ (kPa) (@ 25 °C) | 1.91 | 15.1 | 1.4 | <1 |
| Toxicity | High | Moderate | Low | Low |

ADN (ammonium dinitramide)-based green propellants development started at the Swedish Defense Research Agency (FOI) in Europe in 1997 [22,50,51]. The ADN-based monopropellants family mainly consists of FLP-103, 105, 106, 107 and LMP-103S, where the latter was developed by Bradford ECAPS Co. LMP-103S and FLP-106 are the most mature, and the former was qualified by the European Space Agency (ESA) and in-space demonstrated through the High Performance Green Propulsion system (HPGP) on the Mango-PRISMA satellite launched in June 2010 [52–54]. Different fuels were used within this energetic ionic liquid mixture such as methanol, monomethyl-formamide MMF and dimethyl-formamide DMF. However, methanol was found incompatible with ADN unless by addition of ammonia (NH$_3$) in order to increase the pH of the mixture [22,55]. Composition of some ADN-based monopropellants are shown in Table 5 where the performance of the FLP-family is shown to be higher than LMP-103S. However, all ADN-based monopropellants mentioned in this study possess volumetric specific impulse lower than that of AF-M315E (391 g s cm$^{-3}$). ADN-based green monopropellants are not only ignited by preheated catalytic beds, same as all monopropellants, but can also be ignited electrically or using thermal ignition. Larsson et al. [56] found that ADN-based propellants can be ignited using resistive heating by conducting electrical current through the propellants, and very rapid ignition was obtained (less than 2 ms); moreover, the least amount of electric energy utilized for successful ignition was in terms of (20 J). While Wilhelm et al. [53] found that glow-plug ignition was successful for LMP-103S and FLP-106, satisfying ignition behavior and decomposition. Advantages of the LMP-103S and FLP-family over AF-M315E include, but are not limited to, lower combustion temperature, which allows using materials with lower melting point, and simpler designs for thruster development. Moreover, flexibility in using different ignition techniques and not just being restricted to catalytic decomposition of ADN-based green monopropellants would allow for development of novel designs of monopropellant thrusters.

**Table 5.** ADN-based monopropellants properties [22,57,58] (ideal vacuum $I_{sp}$ by [57] using NASA CEA @ 2.0 MPa chamber pressure, 50:1 expansion ratio assuming frozen condition [53]).

| Propellant. | Formulation | Theoretical $I_{sp}$ (s) | Density (g cm$^{-3}$) * | $\rho I_{sp}$ (g s cm$^{-3}$) | $T_c$ (°C) |
|---|---|---|---|---|---|
| LMP-103S | [1] 63.0% [2] 18.4% [6] 18.6% | 252 | 1.24 | 312.48 | 1630 |
| FLP-103 | [1] 63.4% [2] 11.2% [5] 25.4% | 254 | 1.31 | 332.74 | 1760 |
| FLP-106 | [1] 64.6% [3] 11.5% [5] 23.9% | 255 | 1.357 | 344.6 | 1814 |
| FLP-107 | [1] 65.4% [4] 9.3% [5] 25.3% | 258 | 1.351 | 348.5 | 1869 |

[1] ADN. [2] Methanol. [3] MMF. [4] DMF. [5] Water. [6] Ammonia (aq. 25% concentration). * @ 20 °C.

### 2.2. Liquid NOx Monopropellants

In this section green monopropellants of N$_2$O, nitro compounds, and premixed N$_2$O with hydrocarbons (i.e., nitrous oxide fuel blends NOFB [44,59]) will be introduced, emphasizing on relevant properties needed for propulsion system design either for small satellites or high-thrust in-space propulsion up to nearly 600 N.

Generally, the binary compounds of nitrogen and oxygen (e.g., NO, NO$_2$, N$_2$O, N$_2$O$_3$, N$_2$O$_4$) have been considered as oxidizers in hybrid rocket engines and bipropellant sys-

tems [60]. Nitric oxide NO is gaseous in the typically requested temperature-pressure range of [−30, +80] °C and [0.1, 3] MPa, which does not suit the propulsion applications of concern, since the sought propellant is required to be in liquid phase and easily storable in lighter weight tanks within the given operation conditions. Although $NO_2$, $N_2O_3$, and $N_2O_4$ form an equilibrium mixture within the mentioned temperature–pressure range, they are still highly toxic and considered as GHS class 1 acute-toxicity [6]. Nitrous oxide $N_2O$ is the only compound in this group that falls under green propellant umbrella since it is considered GHS class 5 relatively nontoxic, moreover, the critical point stands at 36.4 °C and 7.24 MPa [61] and is liquid in a wide part of the pressure–temperature range mentioned above. $N_2O$ possesses good storability characteristics at room temperature especially for long term storage since it does not have decomposition or boiling problems when compared to $H_2O_2$ or cryogenic LOX as examples. At 20 °C the saturated vapor pressure of nitrous oxide is ~5.2 MPa, which is high compared to EILs and HPAS, but still considered favorable when considering this propellant for self-pressurizing feed systems.

$N_2O$ (liquid) storage density is ~0.745 g cm$^{-3}$ at 20 °C and ~5.2 MPa vapor pressure [62]. Thus, with suitable pressure vessels $N_2O$ can be kept under stable and readily-operating conditions, adding to that, its material compatibility with common tank materials including metals, plastics and composite-materials [60]. Although $N_2O$ as monopropellant has lower performance than most EILs, it has an experimental result $I_{sp}$ = 206 s ($T_c$ = 1913.15 K, @ $p_c$ = 0.3 MPa, nozzle expansion 200:1 [63]), which is higher than high-test peroxide HTP ~180 s [62]. The most compelling about $N_2O$ for modern propulsion system design is that it can be used in the so called "multi-mode" propulsion system [48], where it can act as a propellant for cold-gas, monopropellant propulsion, and/or bipropellant systems while sharing the same propellant tank.

Nitromethane (NM, $CH_3NO_2$) is a promising nitro compound green monopropellant candidate for modern in-space propulsion systems for relatively low- to high-thrust range. NM is a relatively nontoxic (GHS class 4 toxicity), viscous, flammable liquid with density of 1.1371 g cm$^{-3}$, and has a freezing temperature of −28.4 °C [64]. It shows good storability for in-space applications and by adding stabilizer additives (such as, ditertiarybutyl peroxide or chloral, and diacetyl [65]) it could be a highly-attractive liquid monopropellant [66]. Nitromethane can be considered in "multi-mode" systems since it can be used in both mono- and bipropellant propulsion systems, thus increase the overall system optimization. NM as a green monopropellant, possesses high performance ($I_{sp}$ = 289 s) [3], high volumetric specific impulse, and combustion temperature around 2449 K.

NOFB nitrous oxide fuel blends were studied as monopropellants since World War II by the Germans [67]; however, their utilization and development were ceased due to the availability of hydrazine since the 1960s until recently. Because of the current economic and political reforms to greenify the use of propellants, nitrous oxide fuel blends research has been revived in Europe by the European Fuel Blend Development (EUFBD) program carried between 2015–2018 [68].

NOFBX$^{TM}$ monopropellant was invented in the U.S. between 2005 and 2007 under the NASA Mars Advanced Technology program, and later a proprietary of Firestar company [69]. NOFBX$^{TM}$ was demonstrated in the 0.4 N–445 N thrust range with measured specific impulse performance around 325 s, while the theoretical value was ~345 s and chamber temperature ~3200 K at 0.7 MPa with stoichiometric O/F = 3 (or 25%fuel). It is storable as saturated liquid under a wide range of temperatures, with a critical point of 39.48 °C and 7.19 MPa [70].

HyNOx or Hydrocarbon NOx as denoted by German scientists in the German Aerospace Center (DLR) are premixed monopropellants where oxidizer and fuel are premixed in a single tank [71]. HyNOx fuel blend used nitrous oxide with ethylene (i.e., ethene IUPAC name) ($N_2O/C_2H_4$) due to the similar vapor pressure of the two compounds [72] and has density of 0.879 g cm$^{-3}$. The recorded theoretical vacuum specific impulse is 303 s obtained with O/F ratio = 6 (~14.29%fuel) under 230 K and 2.5 MPa temperature and

pressure [44]. A high combustion temperature up to 3264 K is also recorded as a drawback besides the important need for a flashback-arrestor, by Werling et al. [73].

Nitrous oxide/ethanol ($N_2O$/$C_2H_5OH$), another hydrocarbon NOx blend, was the mixture of choice after a tradeoff study carried by Mayer et al. [74] in the EUFBD program. Among hydrocarbons, ethanol showed better ignitibility and moderate flame temperature, and further demonstrates stability and miscibility with nitrous oxide. Physical properties of the selected mixture resulted in saturated liquid density of 0.892 g cm$^{-3}$ with a stoichiometric O/F ratio of 5.73 (~14.86%fuel), the critical point stood at 36.45 °C and 6.3 MPa and the vapor pressure at bubble point was 2.6 MPa, while a theoretical specific impulse $I_{sp}$ = 331 s and combustion temperature 3093 K were reported [74]. A test campaign was carried out with a 600 N thruster and specific impulse of 259 s was achievable [75]. Drawbacks reported during this study are: high combustion temperature, which requires complex engine design and active cooling system, incompatibility of the nitrous oxide fuel blend with titanium was expected to exist, possibility of flammable vapors in the propellant tank, and low density at practical storage temperatures [72,74,75]. Advantages of nitrous oxide fuel blends, as most green monopropellants, are nontoxic and noncarcinogenic nature, low freezing point, higher specific impulse than hydrazine, and the most prominent advantage is self-pressurization capabilities, which allows for simple feed-system and tank-pressurization system design. Performance and properties of liquid NOx monopropellants are shown in Table 6.

**Table 6.** Ideal (vacuum) performance and physical properties of the Liquid NOx Monopropellants class (compounds and premixed fuel blends).

| Propellant | Theoretical $I_{sp}$ (s) | $\rho$ (g cm$^{-3}$) [a] | $\rho I_{sp}$ (g s cm$^{-3}$) | $T_c$ (°C) |
|---|---|---|---|---|
| $N_2O$ (liquid) * | 206 | 0.745 | 153.5 | 1640 |
| Nitromethane ** | 289 | 1.1371 | 328.6 | 2175.85 |
| NOFBX$^{TM}$ *** | 350 | 0.700 | 245 | 2926.85 |
| HyNOx (Ethene) [†] | 303 | 0.879 | 266.3 | 2990.85 |
| NOx/Ethanol [‡] | 331 | 0.892 | 295.3 | 2819.85 |

[a] @ 20 °C. * @ $p_c$ = 0.3 MPa. ** @ $p_c$ =1 MPa. *** @ $p_c$ = 0.7 MPa and stoichiometric O/F = 3. [†] @ $p_c$ = 2.5 MPa and stoichiometric O/F = 6. [‡] @ $p_c$ =1 MPa and stoichiometric O/F = 5.73.

### 2.3. Hydrogen Peroxide Aqueous Solutions (HPAS)

Hydrogen peroxide ($H_2O_2$) has been used as monopropellant in different aerospace applications since 1938 [76]. $H_2O_2$ is type-classified according to its concentration in aqueous solution, and grade-classified according to the concentration of stabilizers and impurities [77], as shown in Table 7. HTP (High-Test Peroxide) is a highly concentrated $H_2O_2$, greater than >85% weight concentration. Rocket grade HTP is used in space propulsion for low and medium thrust applications and is typically of 98% concentration [78]. The high density of 98% HTP (~1.43 g cm$^{-3}$), and the nontoxic nature, makes it an interesting candidate for storage in propulsion systems in general. In monopropellant systems it can catalytically decompose reaching temperatures in terms of 1222 K [78]. Performance of HTP 98% in monopropellant systems is ~20% less than hydrazine [79], with $I_{sp}$~186 s (at 1 MPa and 50:1 expansion conditions). However, in bipropellant systems, especially with hydrocarbons such as ethanol, it can reach $I_{sp}$ > 325 s (combustion temperature 2752 K) [3], which makes it a very competitive propellant for this kind of propulsion system. Another concentration of hydrogen peroxide that is widely used is the HTP 87.5% with a density of ~1.38 g cm$^{-3}$ at 20 °C and possesses a theoretical specific impulse of ~144 s when evaluated at a chamber pressure of 1 MPa and $A_e/A_t$ of 7.841 at sea level [80]. $H_2O_2$ at concentrations of 90% and 85% were simulated on RPA to give, respectively, theoretical vacuum specific impulse of 172.13 and 150.47 s, chamber temperature 1019.3 and 892.65 K at 1 MPa, and expansion ratios of 40:1 and 10:1 applying the shifting equilibrium model for the whole nozzle.

The Hydrogen Peroxide Aqueous Solutions (HPAS) class possesses the lowest performance values among green monopropellants; however, a unique characteristic of this family of propellants can make it of high interest from the point of view of rocket propulsion designers in terms of increasing the overall system performance and size optimization. This unique characteristic is the hypergolic ignition of hydrogen peroxide with various propellants making it a distinguishable candidate for in-space propulsion systems. Hydrogen peroxide has been experimented thoroughly for hypergolic ignition with hydrocarbons such as ethanol [81] and propyne [82]. Further, hypergolic ignition with ionic-liquid fuels such as 1-ethyl-3-methyl imidazolium cyanoborohydride ([EMIM][BH$_3$CN]) [83] and 1-allyl-3-ethyl imidazolium cyanoborohydride ([AEIM][BH$_3$CN]) [84] would allow for developing new generations of green propellants for effective in-space bipropellant propulsion systems, namely green hypergolic ionic liquids (HILs) [85].

Undoubtedly, the utilization of hydrogen peroxide as a well-studied and experimented propellant in modern green propulsion systems can be of great benefits to the overall system optimization and overall increase in performance, since it would allow for implementation of multi-mode [49] systems where two or more propulsion systems can rely on the same propellant tank. In a multi-mode system, hydrogen peroxide can decompose catalytically in monopropellant auxiliary propulsion (e.g., reaction control system RCS or attitude control system ACS), as well as being used as an oxidizer in bipropellant primary propulsion system as it distinguishably can ignite through a hypergolic reaction without the need for separate ignition power source. A recent research work by Rhodes and Ronney (2019, 2020) theorized [86], investigated, and experimented [87] a first prototype of a *Hydrogen Peroxide Vapor Propulsion*, which can provide attitude control capability for small satellites with thrust range of millinewtons. Although hydrogen peroxide in vapor phase is extremely unstable and prone to detonation as widely known, in this novel concept it is claimed that the reactive vapor phase was utilized within a low-thrust propulsion system where the vapor was used as a propellant. Hydrogen peroxide reactive vapor was vacuum evaporated from the surface of the stored propellant in liquid phase and then passed over a catalytic bed where a chemical reaction occurs and temperature is increased; this eventually leads to a theoretical specific impulse in vacuum (>200 s), which is very high when compared to conventional H$_2$O$_2$ systems. This proposed novel system would add to the scope of applications of hydrogen peroxide aqueous solution propellants.

**Table 7.** Physical and chemical properties of H$_2$O$_2$ propellant with different concentrations [77,88].

| Properties | H$_2$O$_2$ Propellant Classification | | | | |
|---|---|---|---|---|---|
| | Type 70 Grade ES | Type 85 Grade ES | Type 90 Grade ES | Grade HP | Type 98 Grade HP |
| Concentration % | 71.0–73.0 | 85.0–87.0 | 90.0–91.5 | | 98.0–99.0 |
| Stability * | | | ≤2% | | |
| Density $\rho$ (g cm$^{-3}$) ** | ~1.29 | ~1.34 | ~1.40 | | ~1.43 |
| Freezing point (°C) | −40 | −17 | −12 | | −2 |
| Boiling Point (°C) | 125 | 137 | 140 | | 147 |

ES: Extra Stabilizers; HP: High Purity; * (24 h/100 °C) %Loss of active O$_2$; ** @ 20 °C.

## 3. Green Monopropellants in Multi-Mode Propulsion

This section aims at introducing some applications of the discussed green monopropellants in different propulsion systems. Since the choice of a monopropellant propulsion system is affected by the overall system requirements of the spacecraft, depending on different missions and applications, it was important to shed light on different integration possibilities of monopropellant propulsion in various propulsion system configurations and layouts. Definitions of terms describing different propulsion system configurations (such as multi-mode, dual propulsion, and combined chemical-electric propulsion) are highlighted from the respective literature and explained in order to clarify any ambiguities in using such terms and definitions.

Multi-mode propulsion systems, as described briefly earlier, are capable of utilizing the same propellant tank for different types of propulsion at the same time. This technique will optimize the size and allowable volume inside a spacecraft and of particular importance for in-space systems thriving for miniaturization and performance increase. Two or more propulsion systems such as cold-gas, monopropellant, bipropellant, and electric propulsion can work alongside utilizing fewer propellant tanks. Figure 1 shows four types of multi-mode propulsion systems utilizing monopropellants. The CubeSat propulsion system schematic shown in Figure 1a is an example of EILs green monopropellant system. The Modular Impulsive Green Monopropellant Propulsion System (MIMPS-G) [89] operates a primary catalytically decomposed monopropellant propulsion that is characterized by high-thrust impulsive capabilities. Auxiliary vapor propulsion system extends from the same propellant tank to provide reaction and attitude control for the spacecraft. The system miniaturization is positively impacted by the use of only one propellant tank alongside a miniaturized autogenous feed and pressurization system of a micro electric pump-feed cycle.

Nitrous oxide as discussed in Section 2.2 is a widely studied green propellant for its ability to perform in different types of propulsion systems. Already available commercial CubeSat propulsion modules use $N_2O$ and hydrocarbon fuel (propene) in bipropellant primary propulsion systems [90]. Other studies investigated the use of RP-1, ethylene, and ethanol fuels with $N_2O$ as an oxidizer for bipropellant propulsion, along with the ability to utilize the $N_2O$ for cold-gas and/or monopropellant propulsion for multi-mode configuration [60]—Figure 1b shows an example of such system.

$H_2O_2$ as a monopropellant and its ability to ignite hypergolically with green Hypergolic Ionic Liquids (HILs) in a bipropellant system is a relatively new topic that opens many opportunities for research and development of such a promising concept. A recently published article by Korea Advanced Institute of Science and Technology (KAIST) discussed such system using additive-promoters to promote the hypergolic ignition of HIL with 95% $H_2O_2$ [85]. Obviously in such system, an external catalytic bed is no longer needed; however, in the case of extending the HIL/$H_2O_2$ system to a multi-mode configuration incorporating an $H_2O_2$ monopropellant auxiliary propulsion, the need for a catalytic bed will be essential—Figure 1c represents such a multi-mode schematic. Hydrocarbons with hydrogen peroxide bipropellant systems are widely investigated, as mentioned in Section 2.3, and Figure 1d schematizes a multi-mode configuration for such green bipropellant primary propulsion with an auxiliary monopropellant system for reaction and attitude control.

Combined chemical–electric propulsion systems are widely proposed nowadays especially for long interplanetary missions that require high-thrust-low-thrust propulsion capabilities to be able to fulfill efficiently the orbital maneuver requirements of such long missions. This combined chemical-electric propulsion can exist in a "multi-mode" configuration, for different types of electric propulsion thrusters beside the chemical monopropellant thrusters, utilizing a shared propellant tank as J. L. Rovey et al. [49] extensively elaborated. A new concept of Liquid Pulsed Plasma Thruster (LPPT) was being developed by DSSP Aerospace Company [47] that utilizes liquid green electrical monopropellant GEM, discussed in Section 2.1 and is proposed to be a superior alternative to AF-M315E, which may be a game-changer in chemical-electric multi-mode propulsion systems, such as the one shown in Figure 2a, to be designed for such high-thrust-low-thrust long interplanetary missions especially in CubeSats due to their inherited size restrictions.

"Dual propulsion" is another configuration for the combined chemical-electric propulsion systems as proposed by Mani et al. [91,92], which differs from the multi-mode configuration by utilizing two separate systems, a chemical system and another electric propulsion system, alongside each other in the spacecraft. The dual (combined chemical-electric) propulsion system designed by Mani et al. consisted of a green monopropellant regulated pressure-fed system burning the ADN-based FLP-106 green monopropellant, and an

electric propulsion RF ion thruster fueled by iodine. A schematic diagram visualizes this system in Figure 2b.

Finally, green micro-resistojets were being developed in recent years at the Delft University of Technology to operate on liquid water as fuel and to provide thrust in the range of millinewtons in different propulsion applications for micro, nano, and picosatellites—namely CubeSats and PocketQubes [93]. Cervone et al. [94,95] designed two micro-resistojet thrusters, the Vaporizing Liquid Micro-resistojet (VLM), which is based on vaporization and heating of pressurized liquid water, and then expansion in a nozzle. The other thruster is the Low Pressure Micro-resistojet (LPM), which is based on heating and acceleration of the water vapor molecules, in simple geometry slots, under a transitional or free molecular flow regime. These two thrusters will be demonstrated together on the Delfi-PQ satellite as part of the development phase of PocketQubes. However, from the author's point of view, future missions of similar spacecraft classes may be designed to use the developed VLM and LPM thrusters to work alongside in a multi-mode propulsion system fueled by liquid water from a shared propellant storage, schematized in Figure 2c.

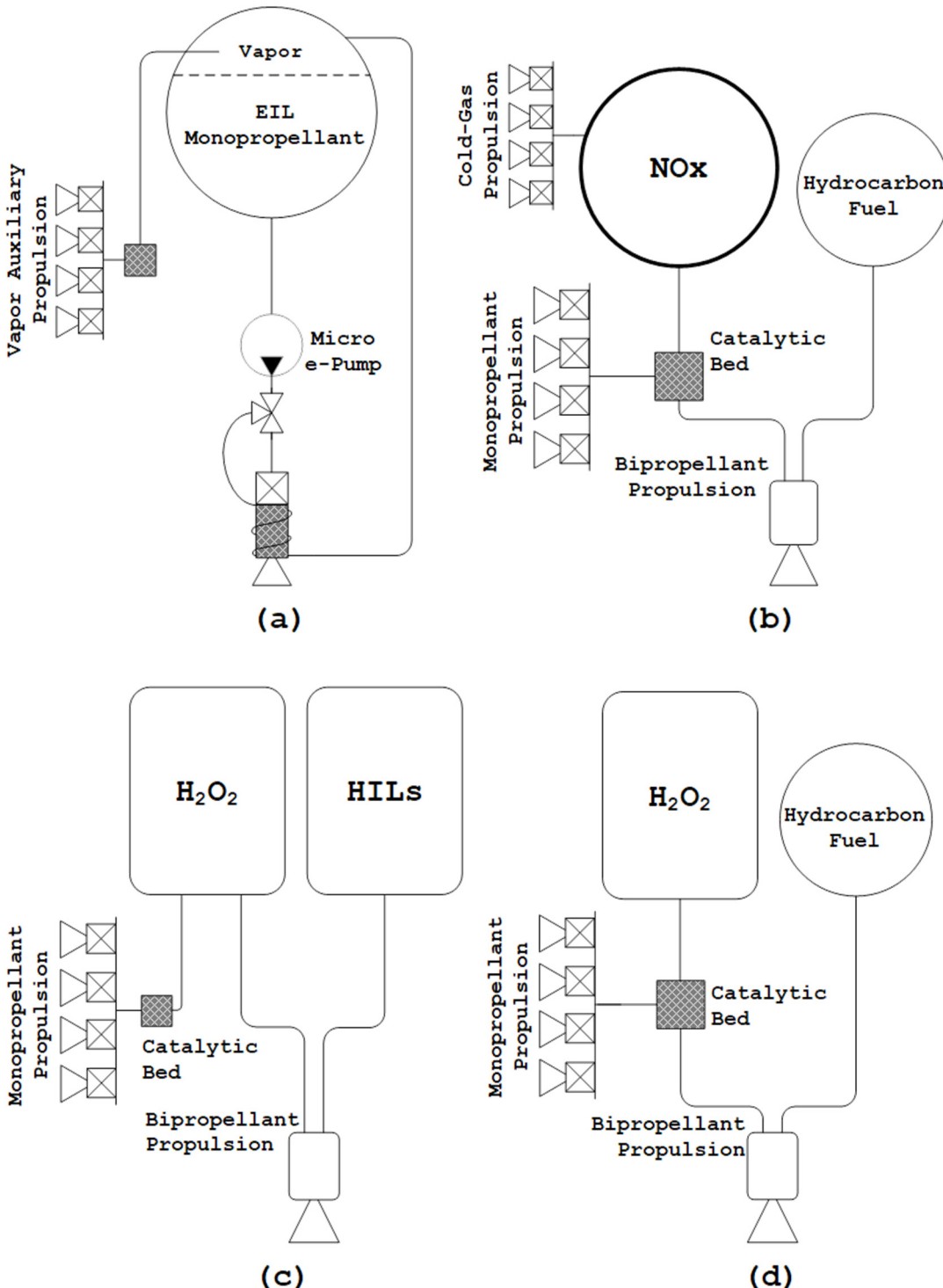

**Figure 1.** Green monopropellants in "multi-mode" propulsion systems: (**a**) Energetic Ionic Liquids (EIL) in micro e-pump-fed autogenously-pressurized primary propulsion system for CubeSat with auxiliary (RCS) vapor propulsion system—(MIMPS-G); (**b**) nitrous oxide with hydrocarbon fuels bipropellant propulsion system with the possibility to incorporate auxiliary propulsion systems of cold-gas or catalytically decomposed monopropellant propulsion; (**c**) ionic liquid fuel hypergolically ignited by hydrogen peroxide bipropellant propulsion system, the $H_2O_2$ can act as fuel for a catalytically decomposed monopropellant auxiliary propulsion system for reaction and attitude control; (**d**) hydrogen peroxide as monopropellant and as oxidizer for hydrocarbon fuels in a hypergolically ignited propulsion system as well as (RCS) in a launcher upper-stage.

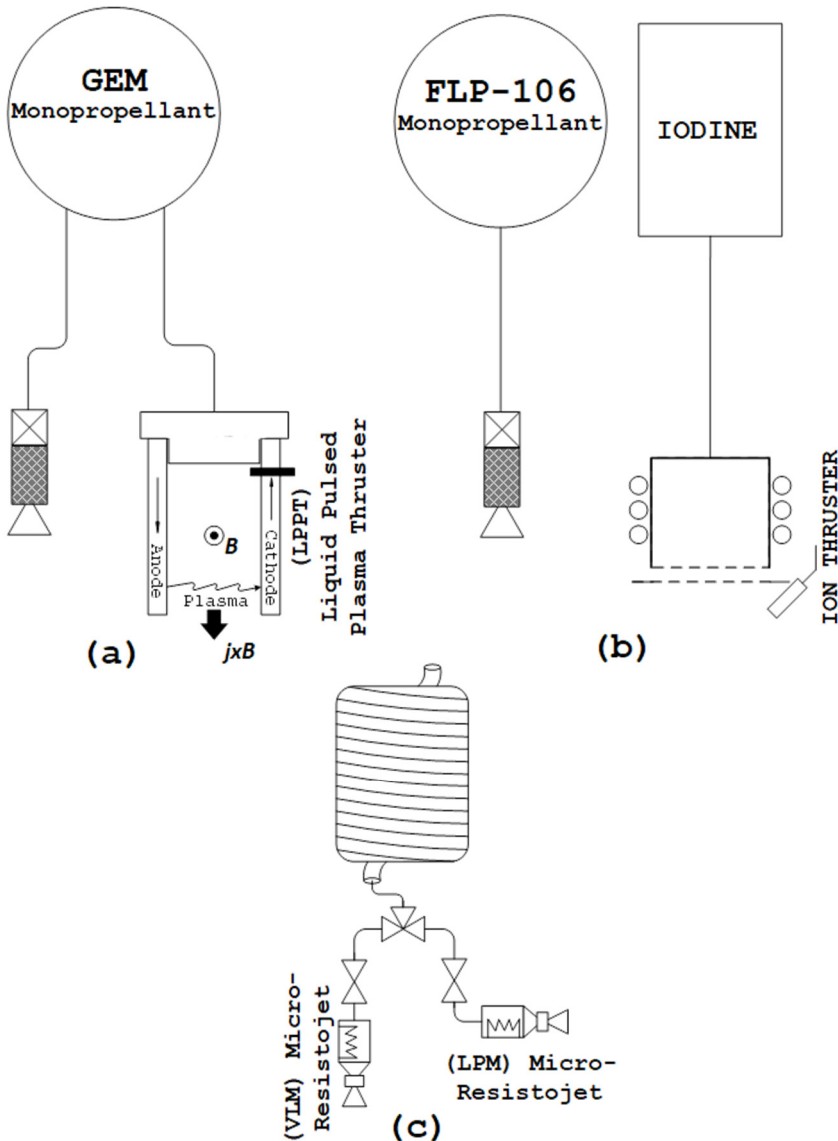

**Figure 2.** Combined Chemical-Electric Propulsion (CCEP): (**a**) a multi-mode configuration of a chemical catalytically decomposed monopropellant system alongside an electric Liquid Pulsed Plasma Thruster (LPPT) both sharing the same Green Electric Monopropellant (GEM) developed by Digital Solid-State Propulsion (DSSP) Aerospace; (**b**) is a CCEP dual propulsion system, the chemical system is burning FLP-106 while the electric propulsion system is separately fueled by iodine. (**c**) A pico-satellite multi-mode electrothermal propulsion system consisting of two micro-resistojets, Vaporizing Liquid Micro-resistojet (VLM) and Low Pressure Micro-resistojet (LPM), both fueled by liquid water as green monopropellant that is provided by a shared storage tank of rolled small diameter tube relying on the capillary action phenomenon [95].

## 4. Green Monopropellants Data and Performance Comparison

In this section collective data on all discussed green monopropellants are tabulated in Table 8. The performance parameters along with the physical and thermochemical properties of any propellant are essential data for preliminary design and assessment for various propulsion systems types taking place in different applications from small-size satellites, CubeSats, deep space spacecraft, to launch vehicles and kick-stages. Analysts and designers of green propulsion systems can find necessary collective data on most state-of-the-art green monopropellants in Table 8. For further details and an in-depth analysis of each propellant, the reader can refer to the relevant subsection in Section 2 of this article

according to the class of the monopropellant of interests. In Table 8, the "conditions" column represents the theoretical evaluation point at the combustion chamber where the monopropellant decomposition was simulated, and these data are referenced to their original authors in each relevant subsection.

 A performance comparison is shown in Figure 3 for fifteen types of green monopropellants compared to hydrazine and following the broad three category classification that was proposed at the beginning of the article. For the Green Electric Monopropellant (GEM) discussed in Section 2.1, theoretical specific impulse is the only parameter noted on the chart since the adiabatic flame temperature was not mentioned in the literature so far, as per the author's knowledge.

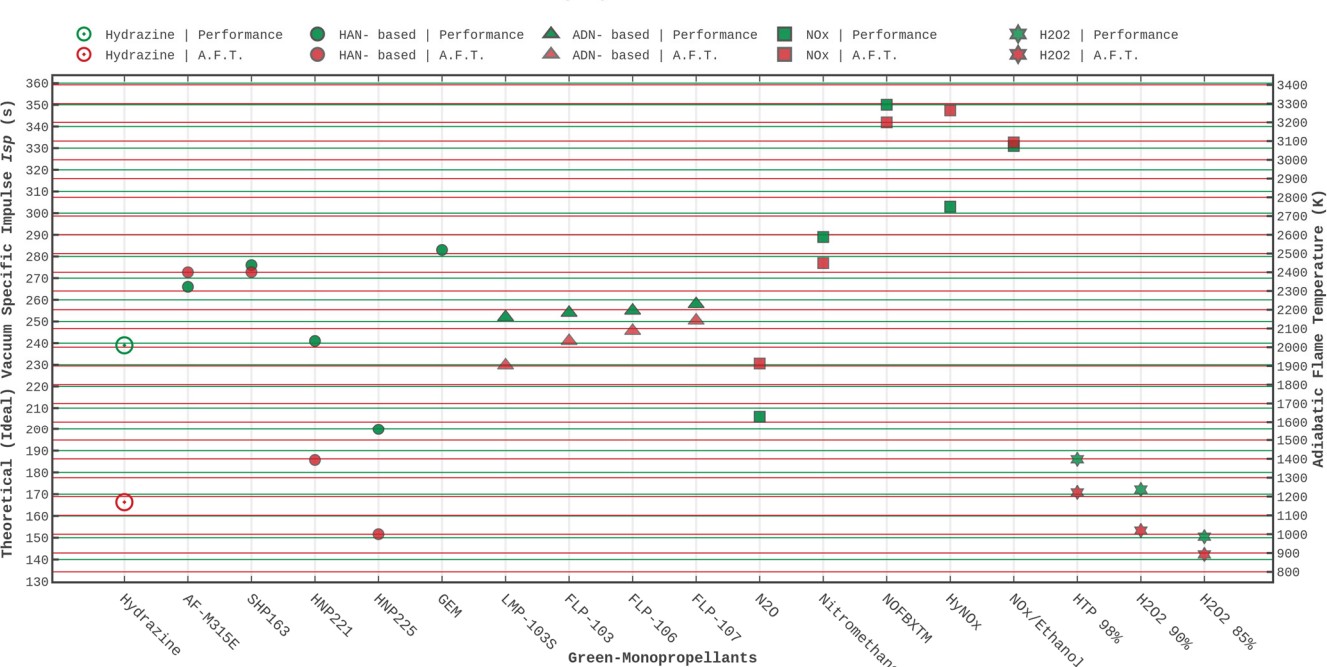

**Figure 3.** State-of-the-art green monopropellants performance chart (hydrazine for reference). The green labels represent the performance (ideal vacuum $I_{sp}$) while the red labels represent the adiabatic flame temperature. Gridlines are colored according to the same convention.

**Table 8.** Performance and physical parameters of state-of-the-art green monopropellants. (Ideal vacuum specific impulse; Density at 20 °C and 1 atm storage conditions).

| Class | Propellant | Theoretical $I_{sp}$ (s) (Vacuum) | Density $\rho$ (g cm$^{-3}$) | Volumetric $\rho I_{sp}$ (g s cm$^{-3}$) | Chamber Temp. $T_c$ (K) | Conditions |
|---|---|---|---|---|---|---|
| | AF-M315E | 266 | 1.47 | 391 | 2166 | 2.0 MPa and $A_e/A_t$ 50:1 |
| | SHP163 | 276 | 1.4 | 386 | 2401 | |
| (EIL) HAN-based | HNP221 | 241 | 1.22 | 294 | 1394 | 1 MPa and $A_e/A_t$ 100:1 |
| | HNP225 | 213 | 1.16 | 245 | 990 | |
| | GEM | 283 | 1.51 | 427 | ? | |
| | LMP-103S | 252 | 1.24 | 312.48 | 1903.15 | |
| (EIL) ADN-based | FLP-103 | 254 | 1.31 | 332.74 | 2033.15 | 2.0 MPa and $A_e/A_t$ 50:1 |
| | FLP-106 | 255 | 1.357 | 344.6 | 2087.15 | |
| | FLP-107 | 258 | 1.351 | 348.5 | 2142.15 | |
| | N$_2$O | 206 | 0.745 | 153.5 | 1913.15 | 0.3 MPa and $A_e/A_t$ 200:1 |
| Liquid NOx Monopropellants | Nitromethane | 289 | 1.1371 | 328.6 | 2449 | 1.0 Mpa and $A_e/A_t$ 50:1 |
| | NOFBX$^{TM}$ | 350 | 0.700 | 245 | 3200 | 0.7 MPa and stoic O/F = 3 |
| | HyNOx (NOx/ethene) | 303 | 0.879 | 266.3 | 3264 | 2.5 MPa and stoic O/F = 6 |
| | NOx/ethanol | 331 | 0.892 | 295.3 | 3093 | 1 MPa and stoic. O/F = 5.73 |
| Hydrogen Peroxide Aqueous Solutions (HPAS) | HTP 98% | 186 | 1.43 | 266 | 1222 | 1 MPa and $A_e/A_t$ 50:1 |
| | H$_2$O$_2$ 90% | 172.13 | 1.39 | 239.3 | 1019.3 | 1 MPa and $A_e/A_t$ 40:1 |
| | H$_2$O$_2$ 85% | 150.5 | 1.37 | 206.2 | 892.65 | 1 MPa and $A_e/A_t$ 10:1 |

## 5. Conclusions

This review article was a result of intensive investigation of state-of-the-art green monopropellants aiming at studying the origins and development status of such propellants to give further insight to the design of novel green propulsion systems for different types of missions. Lately, high $\Delta V$ missions, such as deep space missions and lunar missions, were of rising interest in the space community, and these missions are seeking green monopropellants with novel propulsion system designs to fulfill the demanding orbital requirements. However, it was found that most of the currently available green monopropellants would not only suit deep space missions, but may also be convenient for various applications such as small satellites active orbital operations and upper-stages (kick-stages) of launch vehicles due to their high-thrust impulsive capabilities and favorable physical and thermochemical properties.

Combustion simulation in rocket engine analysis tools, such as RPA and NASA CEA, can be carried out for various EILs using the constituents' thermochemical data provided in Table 1. Most of these data were collected from highly reliable literature between 2018 and 2020 concerned especially with thermodynamic and thermochemical characterization of propellants considered for modern in-space green use. Most non-proprietary propellant formulations, basically the ADN family, were simulated on RPA and results were verified, while the simulated performance parameters and thermodynamic properties were referenced directly to their original authors.

Selection of a particular propellant for a specific application always has one or more driving factors. Considering, for example, the HAN- and ADN-based propellants for application such as impulsive high-thrust demanding orbital maneuvers, the main challenge may be the inherited size restrictions within the structure of the CubeSat, which leads to the need for optimal onboard size utilization as well as the necessity for components miniaturization. Thus, to overcome such restrictions while maintaining high performance, it will be essential, in a general case, to select a propellant with higher volumetric specific impulse. Obviously, that is why propellants such as AF-M315E and LMP-103S are the green monopropellants of choice for such applications where the driving factors are increasing performance and size optimization.

As for nitrous oxide based propellants, the most compelling property is their self-pressurization capabilities. Thus, in propulsion applications where the driving factor is the design simplicity of the feed and pressurization system, NOx fuels are highly sought.

Hydrogen Peroxide Aqueous Solutions (HPAS) inherited a great legacy since the beginning of chemical rockets development, which lies in their maturity, relative operational safety, and high reliability. For applications with less restrictions over size constraints, where the key driving factors are operational safety and high reliability, $H_2O_2$ propellants, such as high concentration HTP, that can both operate in monopropellant propulsion and can hypergolically ignite in bipropellant systems are widely used. One recent—remarkable—application in the field of small-payload reusable launchers that is utilizing hydrogen peroxide aqueous solution HTP in bipropellant propulsion is the Mark-II Aurora spaceplane designed by TU Delft graduates of Dawn Aerospace company [96].

**Author Contributions:** Conceptualization, A.E.S.N., A.C., and A.P.; methodology, investigation, software, data curation, writing—original draft preparation, A.E.S.N.; supervision, writing—review and editing, A.P. and A.C. All authors have read and agreed to the published version of the manuscript.

**Funding:** This research received no external funding.

**Institutional Review Board Statement:** Not applicable.

**Informed Consent Statement:** Not applicable.

**Data Availability Statement:** Any new data created or analyzed was mentioned explicitly within the article and simulation conditions were illustrated.

**Conflicts of Interest:** The authors declare no conflict of interest.

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
