# Peer review of "Review of State-of-the-Art Green Monopropellants: For Propulsion Systems Analysts and Designers"

_aerospace, doi:10.3390/aerospace8010020_

Round 1
Reviewer 1 Report
This is a very welcome addition to the literature. A very comprehensive and well balanced review on green monopropellants.
I recommend acceptance and no need for re-review.
My comments are really minor.
Perhaps the authors could include one paper on recently azide esters:
1. Propellants Explos. Pyrotech. 2019, 44, 1515–1520
Great review !
Author Response
Thank you very much for your comments. Changes were made as requested and details are included in the response letter - please see the attachment.

Reviewer 2 Report
Report on the paper for Aerospace :
“Review of State-of-the-Art Green Monopropellants: for Propulsion Systems Analysts and Designers” 

By Ahmed E. S. Nosseir 1,2,*, Angelo Cervone 1, Angelo Pasini
TO BOTH THE EDITOR AND THE AUTHORS
Authors intend, through a quite important literature collection of ninety-eight references containing adequate data, to propose a review on green monopropellants.
In their review, authors present three families of green monopropellants. A brief history is depicted for each green propellant followed by a description of their development status. A collection of data on properties and performances has also been made and discussed. An interesting comparison of the performances is made at the end of the paper which seems to show the higher performances of NOx. Among the data collected, the enthalpies of formation reported should facilitate the use of analysis tools such as RPA or CEA .
The manuscript is interesting and fits with the scope of the journal. Publishing a collection of data would be a benefit for scientific community, it could help young scientists who need the adequate data to perform performance calculation. However, in my opinion, major improvements have to be done to achieve a complete review.
- The construction of the discussion is clear, except from part 3 about the multi-mode propulsion. The bibliography exposed in this part is interesting but, in my opinion, is not well discussed and introduced even in the title of the work, the abstract and the introduction. Authors must improve their arguments/discussions to demonstrate the pertinence of this part in the proposed review.
- All the references have to be CAREFULLY verified. I have noticed that references are not well given in the manuscript. Example on table 1: for ADN and DMF heat of formation, authors refer to a reference [27] which was written by Wingborg N.. However, the heat of formations given in this study for ADN and DMF respectively come from ref 23 and 16 of the work of Wingborg N.. Authors must always refer to the original work. Another example of issue in the same table (Table 1) is: Why are the references in the legend not the same as the references in the table? What is the purpose of the references given in the legend?
Here, I just gave the example of references given in table 1, but it seems obvious that ALL the work must be verified and corrected.
- All references have to be corrected and written following the journal guidelines. Some authors are missing.
- The originality of the work should be emphasized. Authors should demonstrate what the added value is compared to previous reviews. It seems that some review references are missing. For example the work of Gohardani S. et al. : Green space propulsion: Opportunities and prospects.
- References to research carried out within European projects on green propellants seem to be missing. As an example, the recent Rheform project (H2020) or previously the GRASP project (H2020) could be cited.
- Collection of data could be increased. Figure 3 uses values of table 8 which is a summary of tables 3,4,5 and 6. So they could be summarized but I have the feeling that without table 8 and figure 3 which present the results already exposed in this work, the review suffers a lack of data, especially if the target user is analysts and designers. Performances for each ergol under different conditions could be added. Authors can add performances of H2O2 at different concentrations (e.g. 87.5 wt.%). In table 1, other components could be added. How was the value of heat of formation chosen for each compound (values can differ a bit from a work to another)? It is possible to find different values of ADN heat of formation for example. Giving formulations of HAN-based propellant would also be interesting...
- I noticed numerous typographic and orthographic issues. The paper must be highly improved towards this latter point. There is a lack of homogeneity. Some words are sometimes written with capital letters, sometimes not, the dash is often not well-employed… Tables must be revised. Some mistakes in the legend have to be corrected and the legends should be harmonized between one another. The physical quantities symbol must be in italics and unity must fit the IUPAC standards.
- Abstract is 287 words but should not exceed 200 words (guidelines aerospace)
In addition to the previous general comments, ninety-seven amendment suggestions for more clarity and legibility are presented below. They must be considered and replied (in the case of question). Note that the response to the amendment suggestions will not be sufficient, the review must fulfil the general comments written before.
Following points should be carefully taken into account and/or clarified (“L” refers to the line number written in the left margin of the pdf):
- L13, Remove the dash in “Small-satellites” and write “Small satellites”. Please carefully check all the manuscript in accordance with this modification
- L13, Remove the dash and the space in “nano- satellites” and write “nanosatellites”. Please carefully check all the manuscript in accordance with this modification
- L15, the use of the italics form to underline “important words” seems relatively random in the text. I suggest removing this layout, please carefully check all the manuscript in accordance with this modification. However, physical quantities symbols must be in italics.
- L18, italics and dash must be removed in “green monopropellants”, please write it as followed: “green monopropellant”. Please carefully check all the manuscript in accordance with this modification
- L24, capital letter and dash must be removed in “Green monopropellants”. Please write it as followed: “green monopropellant”. Please carefully check all the manuscript in accordance with this modification
- L25, capital letter must be removed for “Liquid” and “Monopropellant”. Please carefully check all the manuscript in accordance with this modification
- L25, this is an important point which concerns a lot of words in the manuscript. Name of molecules or salts do not take capital letter when they are placed in the middle of a sentence. Please replace “Hydrogen Peroxide” by “hydrogen peroxide”.
- L25, remove the dash in the middle of the sentence and reformulate.
- L31, remove the capital letter for “Hydrogen Peroxide” remove the dash in the middle of the sentence and reformulate
- L33, remove the dash in “Small-satellite”
- L63, remove the capital letter for “Hydrazine”
- L64, please remove “-relatively cool-“. Did you want to write “cold” ? please reformulate the sentence without the use of dashes.
- L65, I suggest removing “at that time”
- L68, remove the capital letter in “Hydrazine”
- L72, remove the dash in “green-propellant”
- L75, I do not clearly understand the meaning of “during various phases of spacecraft development”
- L77, remove the capital letter in “Hydrazine”
- L91, remove the italics and the capital letter in “Nitro Compounds”
- L95, remove the italics and the capital letter in “Nitro Compounds”
- L101, remove the dash in “nitro-compound”
On these two latter points, you can see that there is a lack of homogeneity in spelling and punctuation, and this issue concerns numerous cases. Please check it carefully.
- L104, remove italics for “green” and remove the capital letter for “In-space”
- L106 and 107, remove italics
- L114 and 115 remove italics and dashes for “green-propellants”
- L120 and 122, remove italics.
- L126, rewrite “Green-Monopropellant” as suggested before: “Green monopropellant”
- L139, the point is not discussed in section 2.3. Please modify it.
- L155, remove italics.
- L156, avoid the use of dashes to introduce table, avoid italics
- L160, 163: remove the capital letter in “Methanol”
- L165 please, remove the dash and reformulate the sentence.
- L169, remove the space in “HAN- based”, please write it as followed: “HAN-based”. Please carefully check all the manuscript in accordance with this modification. This is also the case of: ADN-based, HAN/TEAN-based
- L170-173: Please reformulate the sentence: “Different formulations… aqueous solutions”. It seems that punctuation and/or words are missing.
- L179: remove the capital letter in “Hydrazine”
- L184: be careful with the use of capital letters
- L184: Table: g/mol and kJ/mol should be written like this: g mol-1 and kJ mol-1
- L184: Table: Choose between empirical formula and semi-structural formula for molecular fuel. I suggest using semi-structural formula as you did for the DMF.
- L184: Table: Authors must specify in which conditions the heat of formations are given. Standard states?
- Why are the references given in the legend not the same as the references given in the table?
- L187: remove the capital letter in “Green propellant”
- L190: remove the capital letter in “Micro” and “Nano”
- L191-192: Remove italics
- L194: Add “of” after “melting points”
- L196: remove the capital letter in “Hydrazine”
- L201: remove the capital letter in “Methanol” and “Water”
- L201: add a space between values and unities
- L202: change “g/cm3” by “g cm-3” and g.s/cm3” by “g s cm-3”
- L205-207: sentence to be reformulated. The “it is stable” do not fit well in the sentence.
- L209: add “in” before 2019
- L210: remove the space before “based”
- L213: remove the capital letters in “Methanol” and “Water”
- L225: remove the capital letter in “Hydrazine”
- L227: please, move the second parenthesis at the end of the legend or reformulate the legend. There is a space to erase between from and [19]
- L228: Table: change the unity according to the previous comments.
- L229: I am not sure that authors use the right symbol for the “°” in “°C”
- L231: remove the space in “HAN- based”
- 232: remove the capital letters in the molecular name
- L233: remove italics for the company name
- L235: Why do authors use quotation marks in the case of “Multi-mode” ? remove capital letter for multimode. Is a dash necessary in “multimode” ?
- L241 to 243: the meaning of this paragraph could be misinterpreted concerning the paternity of LMP-103s. LMP-103s has been developed by ECAPS. I suggest reformulating this part to make sure that there will not be any misunderstanding.
- L246: add “in” after “launched”
- L248: remove the capital letters in “Methanol” and “Ammonia”
- L249: remove the space in “HAN- based”
- L251: remove the space in “ADN- based”
- L252: change the unity according to the previous comments.
- L252: remove the dash in “green-monopropellants”
- L254: remove the space in “ADN- propellant”, replace “were” by “can be”
- L254-257: reformulate to avoid the repetition of “and”
- L257: et al. must be in italics “et al.”
- L259: remove capital letters in “Physical properties”, replace “state-of-art” by “state-of-the-art”
- L260: Isp must be in italics, please remove the italics form on unities, write “Volumetric specific impulse” before ρIsp. Add the symbol of density and of the vapor pressure in italics.
- L261: Please, harmonize the legend of table 5 following the previous legend and comments. Replace “vac.” by “vacuum”.
- L261: Do not forget the punctuation in the legend
The second half of the manuscript must be carefully corrected following the previous comments. The next comments deal with other issues.
- L263: “include” does not seem to be the appropriate word here
- L266: conclusion of this part should be reformulated since catalytic bed are usually used to simplify and lighten propulsion systems.
- L282: replace “in part of” by “in”
- L291: replace “high-test-peroxide” by “high-test peroxide”
- L296-298: Please, reformulate the sentence “It is…. -28.4 °C [65]”
- L305: replace “its” by “their” (if refers to the blends)
- L306: replace “due” in order to avoid a repetition with the previous sentence
- L325: I suggest erasing “that will provide best performance among candidate hydrocarbons”
- L328-329: It seems that a verb is missing before “[75]”
- L345: I think that a dash in “medium-thrust” is not needed
- L353: Please reformulate without using “-“ between Hydrogen peroxide and aqueous solution
- L360: Avoid the capital letter in molecule name
- L363: remove the dash between green and hypergolic
- L364: the term “as mature” does not seem appropriate here
- L372: Why is the reference (Rhodes & Ronney) not given as the others references? Please, harmonize this point in the entire manuscript.
- L380: add “with different” between propellant and concentrations
- L382: dash
- L383-444: Please, check the use of capital letters and dashes in this entire section (even in the title)
- L398: change the expression “take place”
- L420, 427, 430: references are respectively written this way: J.L Rovey et al. ; Mani et al. ; Mani. Please harmonize. I think that Mani was not alone in the last one. If you need to write “et al.” please write it in italics.
- L421: Add “electrical” between green and monopropellant
- L453: capital letter in not needed for “Class”
- L466: I do not understand the meaning of “however….”
- P521: Figure 3: Isp must be in italics on the axe and in the legend. Please, check the use of capital letters and dashes
Reviewer 3 Report
Dear Authors,
I found this paper valuable as it provides condensed data on green monopropellants. This is a topic very close to my research field. I noticed several points (with minor impact on the quality of the paper) which I kindly request to revise:
Abstract, lines: 16 - 19. The reader may have an impression that green monopropellants provide better performance than bi-propellants. In my opinion this sentence shoud be re-edited. Furthermore, it is not certain for me that storability and stability of green monopropellants is superior over bi-propellants.
Please draw your attention at data given about nitrous oxide:
Lines 93 - 95: N2O is "liquid within temperature-pressure envelope of [-30,+80] oC and [0.1, 3] MPa respectively".
while:
lines 281 - 283: "critical point stands at 36.4 oC and 7.24 MPa [62] which makes it in subcritical state at room temperature and is liquid in part of the pressure-temperature range mentioned above".
Lines 284 - 285: I believe, in case of LOX the problem is boiling rather than decomposition.
Table 8 provides summary data on propulsive performance of selected monopropellants at certain conditions (chamber pressure, expansion ratio). I think that this table would be much more attractive, especially for a propulsion system analysts and designers, if it presented Isp at a single set of conditions.
Moreover, I feel confused about using O/F for monopropellants. As long as the propellant is stored is a single tank, I would suggest to use %fuel rather than O/F to characterize the pre-mix.
Thank you and good luck
Author Response

(The authors gave the same response as above.)

Round 2
Reviewer 2 Report
Second report :
“Review of State-of-the-Art Green Monopropellants: for Propulsion Systems Analysts and Designers” 

By Ahmed E. S. Nosseir 1,2,*, Angelo Cervone 1, Angelo Pasini
TO BOTH THE EDITOR AND THE AUTHORS
I want to thank the authors for their responses, to have considered the points highlighted and for the modifications done on the manuscript.
I have few more suggestions to make:
78: I think that the use of “and being” is not correct here. It seems grammatically not correct. Please change and simplify the sentence.
L162: It seems that “based green monopropellant” does not fit well in the sentence
L208: please fill the first column of table 3 according to table 4. It seems that symbols are missing.
L260: Is a space needed between “FLP-“ and “family” ?
I want to thank again the authors. I think that the content and quality of the review will be a benefit for researches in the field.